# A systematic review of the distribution and prevalence of viruses detected in the *Peromyscus maniculatus* species complex (Rodentia: Cricetidae)

Ally Finkbeiner[1], Ahmad Khatib[2], Nathan Upham[2], Beckett Sterner [2]*

1 Nicholas School of the Environment, Duke University, Durham, North Carolina, United States of America, 2 School of Life Sciences, Arizona State University, Tempe, Arizona, United States of America

* bsterne1@asu.edu

## Abstract

The North American Deermouse, *Peromyscus maniculatus*, is one of the most widespread and abundant mammals on the continent. It is of public health interest as a known host of several viruses that are transmissible to humans and can cause illness, including the acute respiratory disease Hantavirus pulmonary syndrome (HPS). However, recent taxonomic studies indicate that *P. maniculatus* is a complex of multiple species, raising questions about how to identify and interpret three decades of hantavirus monitoring data. We conducted a systematic review investigating the prevalence and spatial distribution of viral taxa detected in wild populations allocated to *P. maniculatus*. From the 49 relevant studies published from 2000 to 2022, we extracted and analyzed spatial occurrence data to calculate weighted populational prevalences for hantaviruses. We found that detection efforts have been concentrated in the Western United States and Mexico with a focus on the spread of Sin Nombre virus (*Orthohantavirus sinnombreense*), the primary causative agent of HPS. There are significant gaps in the existing literature both geographically and regarding the kinds of viruses being sampled. These results are significantly impacted by a recent taxonomic split of *P. maniculatus* into four species, and we were able to update 94% of hantavirus observations to reflect this change. Investigating the uncertain, and likely multiple, phylogenetic histories of these viral hosts should be a key emphasis of future modeling efforts.

## Author summary

Understanding the interactions of viruses with mammal hosts is critical for monitoring disease spread and identifying species or geographic areas at high risk for future zoonotic disease outbreaks. However, much of the data scientists have

**Data availability statement:** The authors confirm that all data underlying the findings are fully available without restriction. All relevant data are within the paper and its Supporting Information files.

**Funding:** The project stemmed from Arizona State University President's Special Initiative Funds to BS. NU was supported by National Institutes of Health grants 5R01GM152813-02 and 1R35GM156919-01. BS was supported by National Institutes of Health grant 1R21AI164268-01/02. The funders had no role in study design, data collection and analysis, decision to publish, or preparation of the manuscript.

**Competing interests:** The authors have declared that no competing interests exist.

collected about viruses in mammals remains fragmented in published literature and is not available in an up-to-date, standardized format online. We conducted a systematic review of virus observations in one of the most abundant, widespread group of rodent species in North America, the *Peromyscus maniculatus* (North American Deermouse) species complex. We extracted and analyzed the spatial distribution of 62,421 observations reporting the results of tests for viruses in the Hantaviridae, Arenaviridae, and Flaviviridae families. We find that conclusions about reservoir host status may be impacted by recent taxonomic proposals that split *P. maniculatus* into four distinct species, and we show how the taxonomic identifications of 94% of the reported observations can be updated using expert-generated maps of the ranges for the proposed new species. We also highlight the uneven geographic sampling of viruses in the rodent populations and major gaps that remain in our knowledge.

## 1. Introduction

This paper presents a systematic review of the existing literature on virus detection studies of wild *P. maniculatus* in order to summarize findings from the last two decades and identify potential areas of further research (Fig 1). The North American Deermouse, *Peromyscus maniculatus* (J. A. Wagner, 1845) is one of the most abundant and widespread mammals native to the continent [1]. The genus *Peromyscus* sec. Mammal Diversity Database v1.12.1 [2–4] includes 82 recognized living species [3] and has been the subject of extensive research in ecology, development, genetics, evolution, and epidemiology [1,5–10]. *P. maniculatus* is of particular interest to the public health sector and has been traditionally well-studied because it is known to be a source of several diseases that are communicable to humans [9,11]. Research has been particularly focused on the North American Deermouse's role as the primary host of Sin Nombre virus (SNV), an RNA virus in the genus *Orthohantavirus* which was first identified in 1993 and was recently renamed *Orthohantavirus sinnombreense* by the International Committee on the Taxonomy of Viruses [12–15]. SNV and other New World hantaviruses are the causative agents of Hantavirus pulmonary syndrome (HPS), an acute respiratory disease with a case fatality rate of 60% at the time of the first outbreaks [8]. From 1993 to 2021, a total of 850 cases of HPS have been reported in the United States [16], with most cases occurring in the Southwest (Fig 2). Despite the relative rarity of HPS cases compared to other hantavirus illnesses, the sudden emergence of the disease and potential for mutation has spurred intensive efforts to characterize the causative viral transmission pathways [18–21]. Modeling the risks of hantavirus disease and novel zoonoses remains an urgent research objective that depends on comprehensive, up-to-date information about pathogen prevalence and the ecological factors influencing disease spread [22,23].

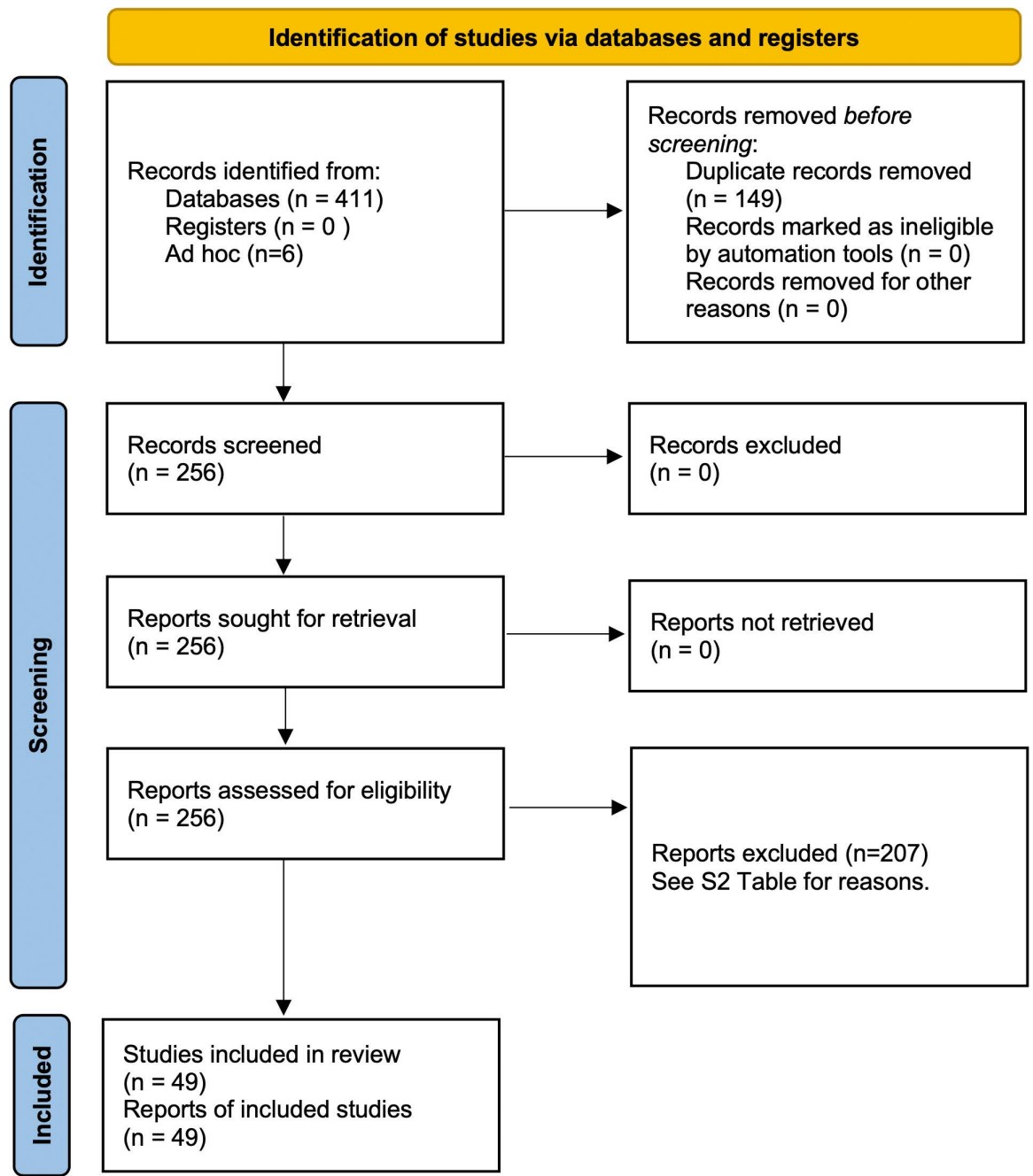

**Fig 1. PRISMA flow diagram.** The flow diagram shows the number of research studies retrieved, included, and excluded in the systematic review process according to the 2020 PRISMA Standard. The template diagram is licensed under CC BY 4.0.

We focus on a spatial analysis of sampling effort and prevalence in studies that tested for hantaviruses and arenaviruses. Our results thereby add to the evidence base for ongoing research into the co-evolution of North American rodents with viral pathogens. In addition, we discuss how study results are relevant for the reservoir host status of *P. maniculatus* considering recently proposed taxonomic revisions.

PLOS Pathogens

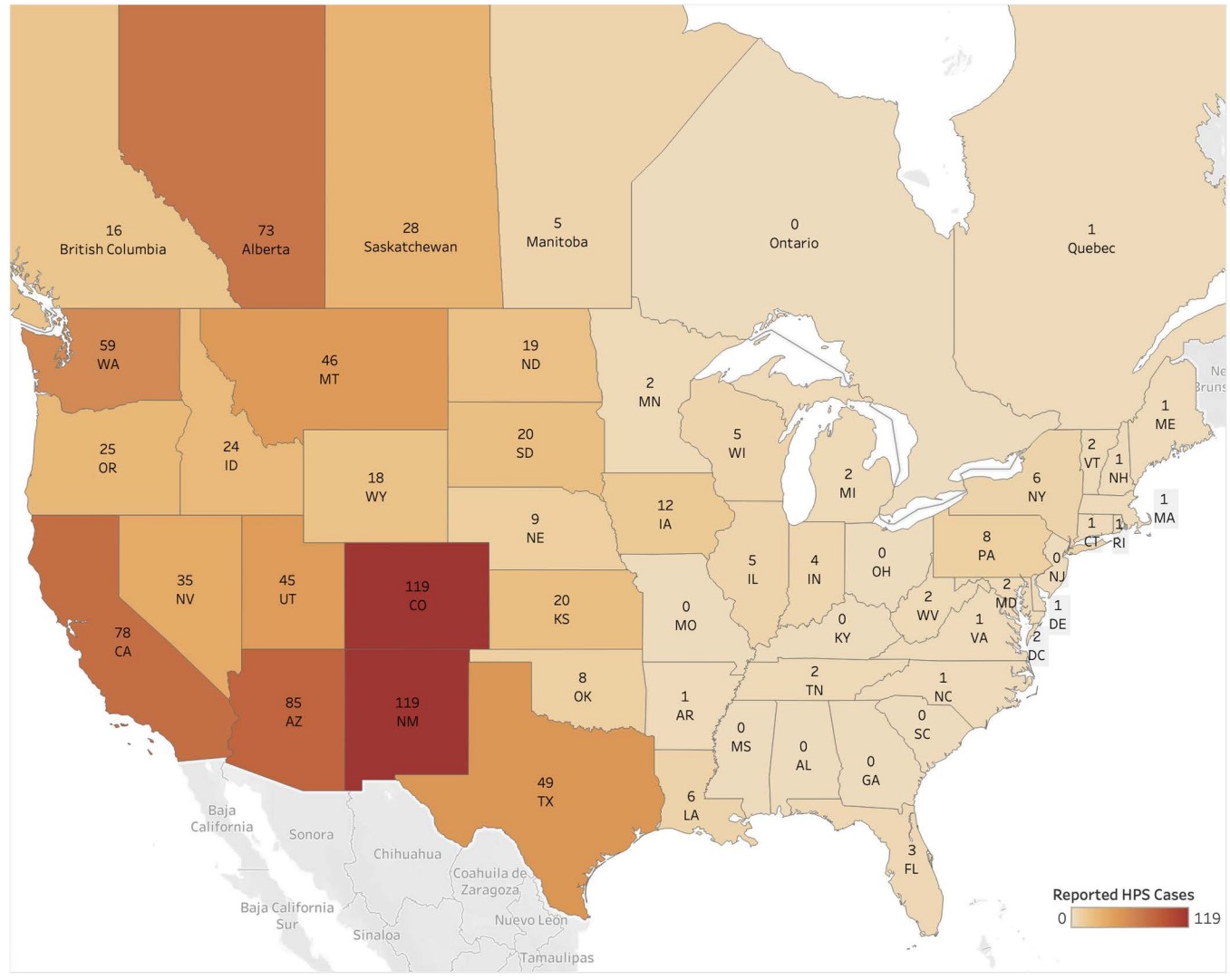

**Fig 2. Map of hantavirus pulmonary syndrome human cases. Cases shown were reported from 1993 to 2021 in the U.S. and to 2020 in Canada** [16,17]. Base map layer from OpenStreetMap under the Open Database License.

Multiple genetic studies indicate *P. maniculatus* is a species complex [24]. It was recently proposed to split *P. maniculatus* into between four and six species-level lineages [2,25,26], though this remains controversial [9]. According to the Mammal Diversity Database v1.12.1 [2–4], the *P. maniculatus* species complex is composed of four species that are distributed as follows: (i) throughout Northern and Central Mexico, referred to *P. labecula* D. G. Elliot, 1903; (ii) in southern California and Baja California, referred to *P. gambelii* (S. F. Baird, 1858); (iii) across the continental United States west of the Mississippi River and into northern Canada, referred to *P. sonoriensis* (Le Conte, 1853); and (iv) east of the Mississippi River until the Atlantic Ocean and north until the Hudson Bay, referred to *P. maniculatus* sensu stricto (J. A. Wagner, 1845) (Fig 3). The genetic distinctiveness of *P. gambelii* and *P. sonoriensis* from each other and from *P. keeni* in the Pacific Northwest [9] supports the existence of multiple species within what has been typically referred to as "*P. maniculatus*" in ecological and biomedical studies. The latter identifications are now

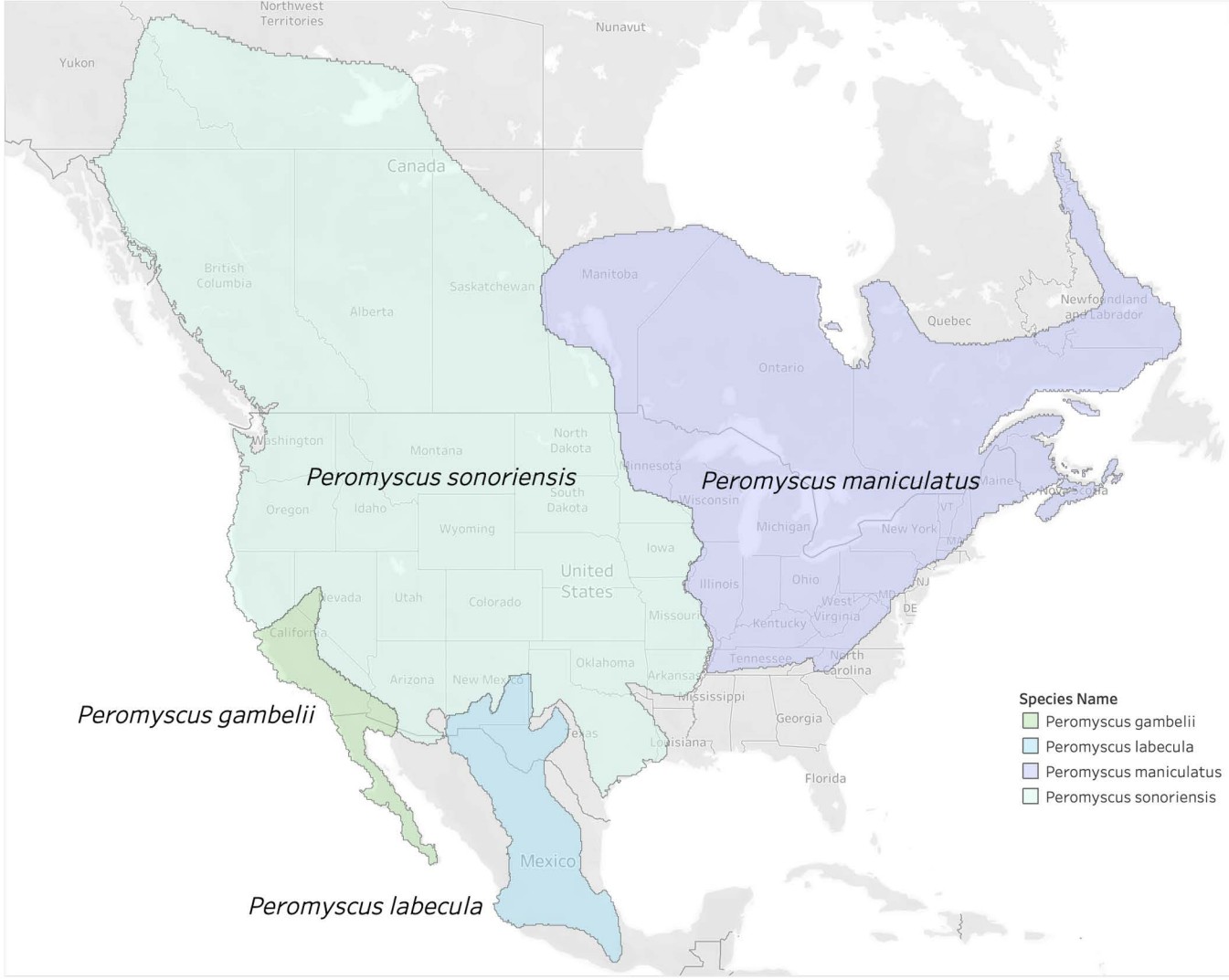

**Fig 3. Proposed taxonomic split of the *Peromyscus maniculatus* species complex.** The proposed split [25,26] divides *P. maniculatus* into four new species as recognized by the Mammal Diversity Database [3]. Note that the name *P. maniculatus* sensu stricto applies east of the Mississippi River in the U.S. and *P. sonoriensis* applies to most of the Southwest where the initial outbreak of HPS was first detected. Base map layer from OpenStreetMap under the Open Database License.

referrable to the continentally distributed *P. maniculatus* sensu lato, and thus require re-interpretation relative to these alternative taxonomic hypotheses.

While hantavirus prevalence is the primary focus of biomedical research on wild-caught *P. maniculatus*, some evidence suggests that rodents in this group may potentially be susceptible to infection by other zoonotic viruses connected with human fatalities, including SARS-CoV-2, flaviviruses associated with tick-borne encephalitis, and arenaviruses [27–29]. Members of this deermouse species complex can be found in every terrestrial ecosystem across the North American continent, including peridomestic areas resulting in frequent contact with humans. Numerous field studies have been conducted on North American Deermice to determine virus prevalence and distribution, especially in regions where out-breaks have previously occurred. These field studies have detected novel viruses and genetic sequencing has allowed

researchers to build up-to-date virus and host phylogenies, but, as we show here, the approach to zoonotic virus monitoring in deermice has not been consistent or standardized.

## 2. Results

Studies have sampled populations in the *P. maniculatus* species complex across 11 different U.S. states, six states in Mexico, and one Canadian province since 2000. Considering 62,663 unique observations across 49 studies, we are able to assign geographic locations to 62,421 (99.6%) at the county (U.S.) or province level (Canada and Mexico). This includes observations that we split evenly across counties when studies only reported aggregate results for two or more counties. While we focus only on observations related to the *P. maniculatus* complex, we note that the full extracted dataset in S3 Table includes observations from over 100 rodent species and that about a third of these had positive test results for an arenavirus, flavivirus or hantavirus. This highlights the taxonomic scope of rodent-virus "dark data" published in scientific articles that is not readily available for open science modeling and research [30].

Scopus returns the largest number of potential papers compared to other databases, regardless of the type of search query used in the literature review (S1 Table). PubMed Central produces more potential papers than PubMed when the search queries are broad, though the latter returns nearly 5 times as many papers when the search query includes virus-related terms as well as host species names. We find that for this particular species, inclusion of common names and junior synonyms in search queries has only marginal effects (<1%) on the total number of citations returned across all three databases. However, this result excludes some additional references that are returned by searching for the common names "deer mice" and "deer mouse," which are not recognized by MDD and are sometimes also used to refer to the whole genus *Peromyscus*. A supplementary search with "deer mice" and "deer mouse" identified three additional papers meeting our inclusion criteria. The small effect of including synonyms on search results may be due to the relatively recent application of alternative species names to populations in the *P. maniculatus* species complex [25,26]. Studies of the influence of including synonyms on search results for other taxa have found larger effects [31,32].

The primary serological method used across all studies is ELISA, though RT-PCR and IFAT, among other tools, are also performed (S2 Table). SNV is the primary target of 40 studies. One study reports results for Monongahela virus, which is now part of SNV according to recent taxonomic updates by the ICTV [33]. Two studies report on the arenavirus species Whitewater Arroyo mammarenavirus (WWAV) and Amapari virus (AMAV), and one study reports on the flavivirus Powassan virus (POWV). Additionally, five studies explicitly sought to detect *Hantaviridae* or *Arenaviridae* antibodies at the family level and do not report results at the virus species level. The seroprevalence ranges greatly across studies from 0% to 100% of individuals sampled in each subpopulation. Sampling effort varies as well, with the smallest study reporting 5 unique observations and the largest reporting 11,391.

We find that interpreting the specificity of reported SNV antibody detection results involves making implicit background assumptions that are not typically stated in the metadata associated with existing species interaction databases. Of the 40 studies reporting antibody results for SNV, 38 use detection methods targeting the nucleocapsid protein on the S segment as the antigen. The remaining two antibody studies [34,35] used a test based on antigen sourced from the lysate of Vero E6 cells infected by Prospect Hill Virus strain PHV1 [36]. Since the nucleocapsid protein is widely conserved among hantaviruses, antibody tests based on this protein in one hantavirus species will also be broadly cross-reactive with other hantaviruses. However, if SNV is the only hantavirus species actually present in *P. maniculatus*, then the antibody tests can still be interpreted as detecting SNV.

To investigate the conclusion that SNV is the only hantavirus species detected in the *P. maniculatus* species complex to date, we analyzed the phylogenetic history of all available hantavirus nucleotide sequences identified with *P. maniculatus* as the host (S1–S3 Figs). For each segment (S, M, and L), almost all hantavirus sequences clustered most closely with the SNV reference sequence compared to other hantavirus reference sequences. The notable exceptions are S segment sequences collected in Texas in 1995 by Rawlings et al. [37] and classified as El Moro Canyon virus (*Orthohantavirus*

*carrizalense*). For all other M and S sequences, bootstrap support of 98 and 86, respectively, supported the clustering of *P. maniculatus*-derived sequences with the SNV reference. For the L segment, a bootstrap value of 100 supports the exclusion of 25 non-SNV reference sequences from the SNV reference and the *P. maniculatus*-derived sequences. The specificity of reported antibody tests is also supported by evidence from studies using PCR tests, which target both the S and M segments and should therefore be sufficient to distinguish SNV from other sequences. In particular, the interpretation of antibody tests as detecting SNV is confirmed by 8 studies that used PCR tests to confirm positive hantavirus antibody results (S2 Table).

With these results in mind, we find that studies reporting SNV results span the longitudinal range from California to Indiana (Fig 4). One or more studies about SNV-*P. maniculatus* interactions has been published in almost every year

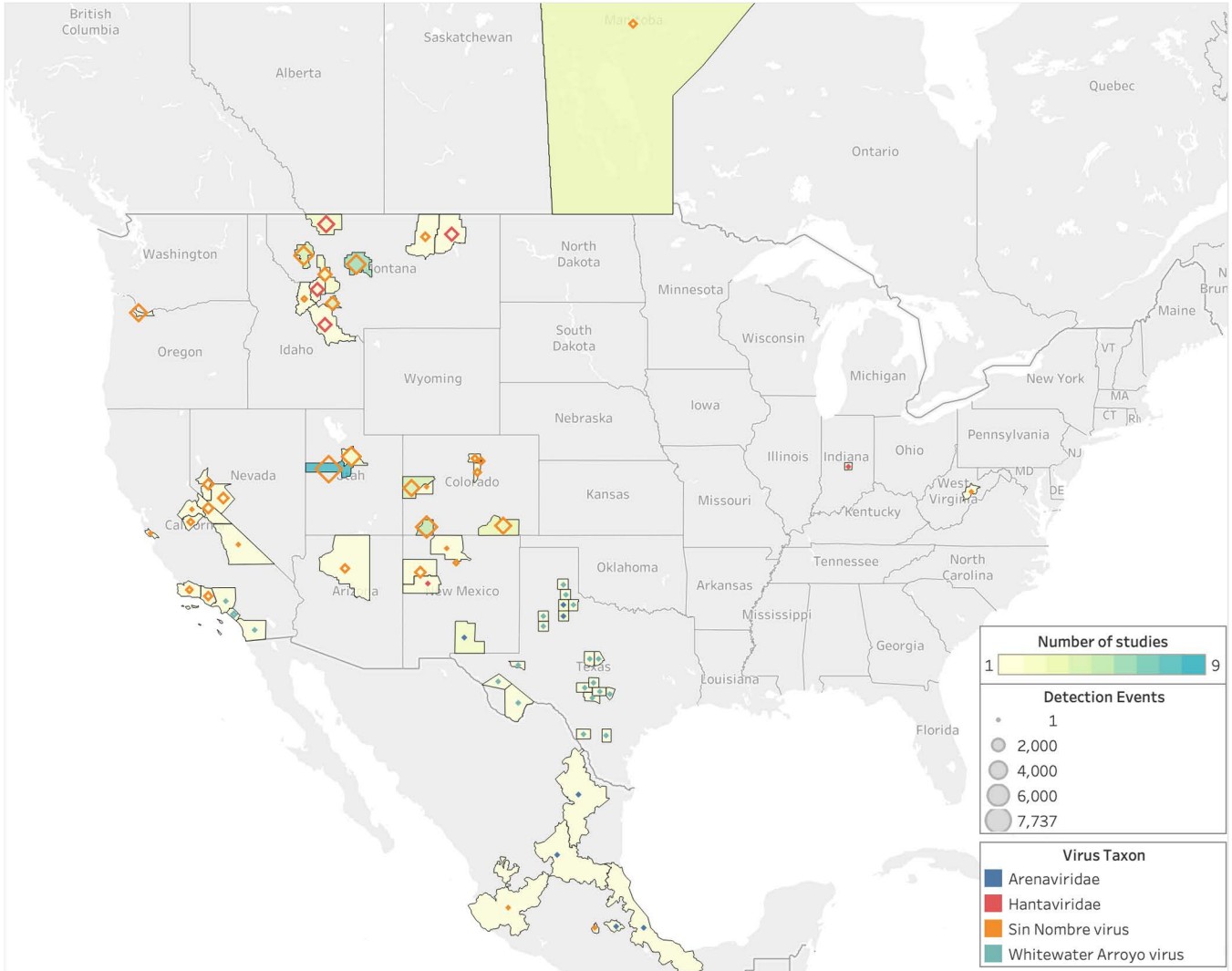

**Fig 4. Geographic distribution of sampling effort.** Sampling effort and zoonotic virus type tested for in *P. maniculatus* field studies. Legend refers to the number of observations in each county, province, or state, where observations are defined as the number of unique hosts tested for a unique virus. Hantavirus and arenavirus refer to studies in which a specific virus species was not identified. Base map layer from OpenStreetMap under the Open Database License.

since 2000. Other viruses are sampled less frequently and at a lower volume, with a maximum number of observations of 353 for both WWAV and AMAV collected in a single study (S2 Table). POWV and arenavirus are investigated in only a single study each. In New Mexico, one study reported finding POWV in *P. maniculatus*, but precise details on location are unavailable [38]. In Texas, multiple studies focused on detecting arenaviruses in *P. maniculatus*, including WWAV and AMAV.

Since 2000, sampling has been concentrated on the western half of the U.S. despite the fact that rodents in the *P. maniculatus* species complex can be found across the entirety of North America (Figs 2 and 3). Counties in Montana, Utah, and Colorado had the highest average sampling effort, with fifteen different counties hosting more than 1,500 unique observations each. This increased level of observations is partially due to multiple studies being carried out in the same county, as some locations are sites of more than one study while others are only sampled once within the last 23 years (S2 Table). In studies that took place over multiple counties, all collection sites are represented. One study is published from Indiana and another from West Virginia, but the rest of the U.S. Midwest and East Coast are not sampled for *P. maniculatus* during the focal interval of 2000–2022.

Sampling effort is more evenly distributed across geographic locations than across virus types. In terms of the number of studies conducted in each state, Montana has twelve while Nevada, Arizona, West Virginia, Oregon, and Indiana each boast one (S2 Table). However, the number of studies alone does not accurately r epresent the total sampling effort in a geographic area; both West Virginia and Oregon only host one study, but the former has a total of 15 observations while the latter has 3,175. Montana has the highest number of observations at 29,857, while Colorado and Utah have the second and third highest counts with 11,794 and 7,889, respectively.

As the number of reported observations increases between 2000 and 2022, so do the number of locations sampled. However, the total number of localities sampled across this time period is only 22 within 11 U.S. states as well as Mexico and Canada, which does not accurately cover the entire range of rodents within the *P. maniculatus* species complex.

The dominance of studies aimed at detecting SNV appears to correlate with historical data on the risk of HPS (Fig 5). There is a moderate positive relationship between the number of reported human HPS cases in a location and the sampling effort in that location (coefficient = 0.06, standard error = 0.008, no intercept; $R^2$ = 0.52; p-value < 0.0001). This trend, however, is driven strongly by a handful of data points. Notably, Montana and Utah are overrepresented in terms of sampling effort compared to the number of historic HPS cases in those locations (studentized residuals of 4.7 and 3.7, respectively). On the other hand, Alberta, Arizona, and Washington are undersampled relative to HPS cases (studentized residuals of -2.2, -2.1, -1.8, respectively).

The highest average seroprevalences are found in New Mexico and California, with Rio Arriba, Cibola, and Inyo counties all reporting greater than 40% of individuals as positive (S2 Table). The majority of counties hovers between seropositivity levels of 10% and 30% of sampled rodents (Fig 6). Several counties in southern California, which fall within the range of *P. gambelii*, find no rodents with virus antibodies among those sampled (S2 Table and Fig 6).

The total number of unique virus species confirmed across all studies is four (S2 Table). The relationship between the number of observations and the number of virus species identified in the literature is not linear. This is because the overwhelming majority of studies are focused on SNV antibodies and thus add to the volume of observations without introducing a novel virus.

Most of the gathered studies are conducted within the range of the proposed species *P. sonoriensis*, which covers most of the western U.S. including the aforementioned states of Montana, Utah, and Colorado (Fig 7). Sampling is also conducted within the ranges of three other putative species within the *P. maniculatus* species complex: *P. gambelii, P.*

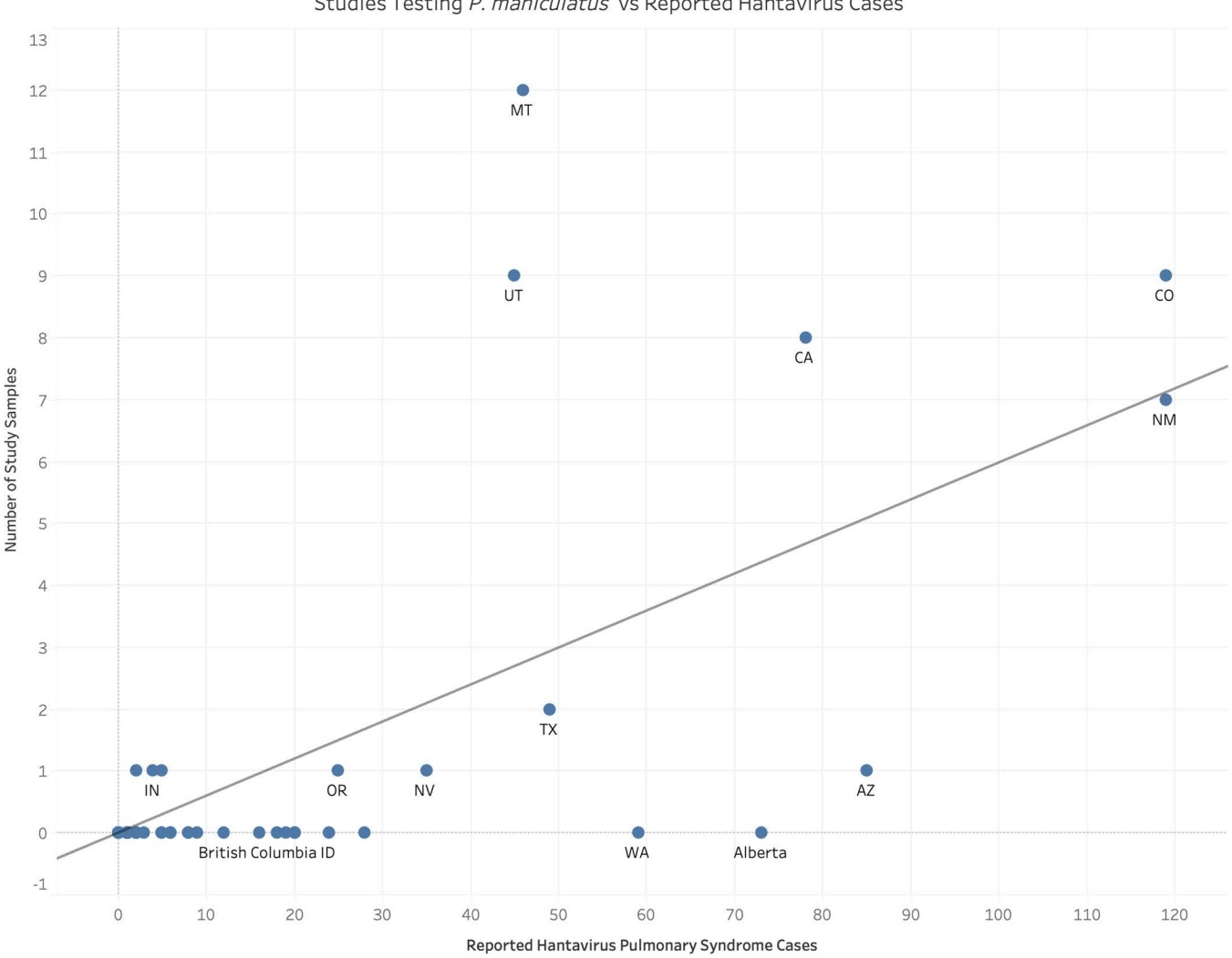

**Fig 5. Hantavirus pulmonary syndrome cases versus number of studies by state.** Comparison of field studies published since 2000 in which testing for viruses in *P. maniculatus* was conducted against the total number of reported human hantavirus cases across states and Canadian provinces.

*labecula,* and *P. maniculatus* (sensu stricto). The arenavirus studies correspond with the range of *P. labecula*, while the SNV and hantavirus studies are concentrated within the range of *P. sonoriensis*.

Based on the non-overlapping expert species range maps (i.e., reciprocally allopatric), we are able to assign 94% (58,845 out of 62,421) of all observations unambiguously to one of the four species (S4 Table). The other 6% are ambiguous with respect to two or more species. We assign an observation unambiguously to a species if the boundaries of the corresponding county or province fall entirely within the range of that species. In some cases, the county or province boundaries fall partly or wholly outside the range of all four species, which we assign unambiguously to a species only if the county or province boundaries overlapped with a single species. Of the 94% of observation that we could assign, *P. sonoriensis* received by far the most assignments (90%) compared to 6% for *P. gambelii*, 2% for *P. maniculatus* sensu stricto, and <1% for *P. labecula*.

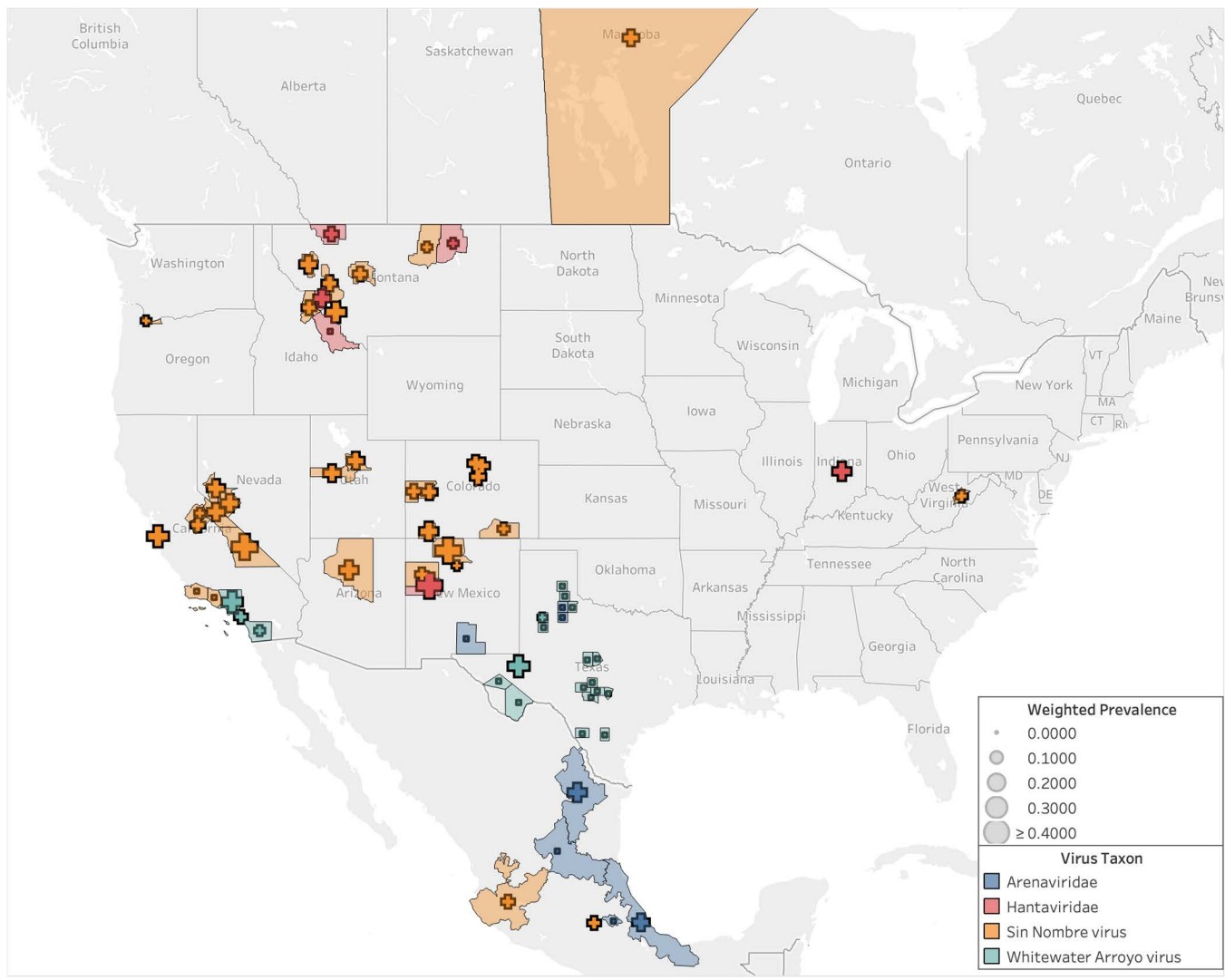

**Fig 6. Map of seroprevalences.** Seroprevalence for viruses studied in *P. maniculatus* by United States county, Canadian province, and Mexican state. In areas in which there was more than one detection study conducted, the average seroprevalence across studies was weighted by sampling effort. Base map layer from OpenStreetMap under the Open Database License.

## 3. Discussion

Zoonotic viruses in the species complex *P. maniculatus* have been non-systematically monitored over the last 20 + years. In scientific literature published since 2000, we found reports of 62,663 unique observations of wild *P. maniculatus* (sensu lato) interacting with four different virus species from three viral families. We also noted a fifth species, El Moro Canyon virus (*Orthohantavirus carrizalense*), which was collected in Texas during a 1995 study outside our review scope. This body of research covers 11 unique U.S. states as well as regions in Mexico and Canada, but it does not reflect the entirety of the range of the *P. maniculatus* species complex, indicating that there are significant gaps in the literature within the 2000 – 2022 period.

Sampling effort varied significantly by both geographical location and virus genotype, with Sin Nombre virus (SNV) overrepresented in the literature relative to other virus families, especially in the states of Montana and Colorado. With

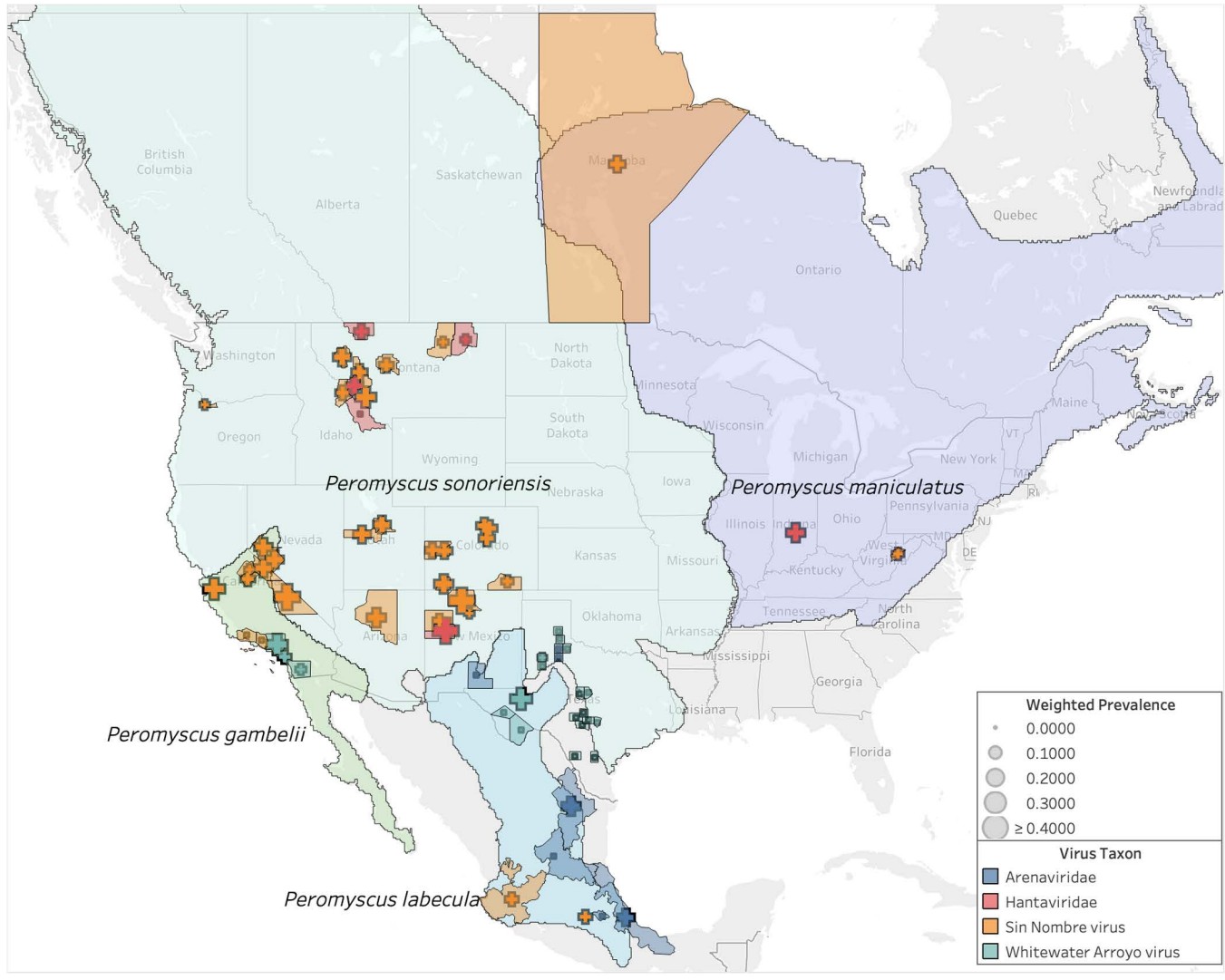

**Fig 7. Effect of taxonomic split on virus data.** Seroprevalence compared to host ranges in the *P. maniculatus* species complex. Base map layer from OpenStreetMap under the Open Database License.

82% of the studies reporting on SNV, the SNV prevalence in *P. maniculatus* over the last 22 years is likely the best representation of hantaviral sharing dynamics in this group of rodents. Results on Powassan virus, Whitewater Arroyo Virus, and Amapari virus are informative in that they indicate the presence of other virus families within the *P. maniculatus* complex. However, further conclusions cannot be drawn about the distribution of these virus families without additional targeted sampling.

This systematic review suggests that, over time, even as more populations referrable to the *P. maniculatus* species complex are being sampled for zoonotic viruses, new viruses are not being discovered. While at first glance this may suggest that virus sampling for *P. maniculatus* is adequate, it more likely indicates that sampling efforts have not been sufficiently dedicated to discovering other known or novel viruses [7]. Instead, researchers have been targeting Sin Nombre virus using antibody and PCR tests given their focus on understanding the public health risks posed by the distribution and prevalence of certain known viruses. Hantaviruses in general were sampled far more than arenaviruses or

flaviviruses, which reflects public health concerns relative to HPS cases. However, both arenaviruses and flaviviruses are known to be capable of spreading to humans [39,40], suggesting that increased sampling effort dedicated to these viral families will provide valuable public health information.

Only 22% of U.S. states were represented in the literature even though the *P. maniculatus* species complex is widespread across the continental U.S. except for the Southeast, where other *Peromyscus* species are found (e.g., *P. polionotus*, *P. leucopus*). While Indiana and West Virginia each received only one study, the rest of the U.S. Midwest and East Coast has remained unsampled for *P. maniculatus*-to-hantavirus interactions despite regional virus detection studies being conducted in other rodent species over the same time period [41–43]. HPS disease cases have been reported only sparingly in states east of the Mississippi since 1993 (Fig 2), so it is unsurprising that recent sampling efforts have not been dedicated to these regions where the risk of disease transmission to humans is sufficiently low (Fig 4). Interestingly, a handful of states where a significant number of HPS have historically been reported—Washington, Wyoming, North Dakota, and South Dakota—were not sampled for *P. maniculatus* occurrences in any of the studies, even though new cases have occurred in each of these places within the last decade [16].

Among the geographic locations that were sampled (Fig 4), most populations fall largely within the subdivided range of *P. sonoriensis*, and the dominant virus over this range was SNV (Fig 7). Surprisingly, only two sampling locations are now associated with *P. maniculatus* sensu stricto: Marion County, Indiana, and Randolph County, West Virginia. The latter was found positive for Monongahela virus, which is now grouped taxonomically with SNV, and the former was a positive for a generic hantavirus ELISA test.

Seroprevalence varied greatly across geographic regions, and the associated variation in sampling effort makes it difficult to draw conclusions about regional trends of virus prevalence. Several counties in Montana, Utah, and Colorado reported seropositivity of greater than 30%, while numerous studies conducted in counties in California, New Mexico, and Texas found zero positive individuals. Since sampling effort was highly inconsistent across all locations, the seropositivity results should be considered as reflections of not only the prevalence of hantavirus in the given region but also of the number of rodents sampled and the time period over which sampling took place. Additionally, many of the studies we included investigated demographic and ecological factors that may influence Sin Nombre prevalence, including rodent age and sex, biotic interactions with other rodent species, and abiotic effects from temperature and rainfall [e.g., 44–46,35]. More consistent and even sampling efforts across the wider range of *P. maniculatus* would be necessary to more conclusively investigate the geographic trends in virus prevalence.

The virus detection dataset we extracted (S3 Table) represents the combined detection results reported by multiple long-term ecological monitoring studies as well as short-term sampling studies. Prior analyses of these studies have explored important environmental and behavioral hypotheses about the likely causes of Sin Nombre outbreaks in humans [e.g., 47–53]. Especially in arid environments such as the desert Southwest, models have shown support for increased precipitation as a key driver of *P. maniculatus* abundance and Sin Nombre prevalence. Ecological interactions with predators and other rodent species may also be important factors influencing abundance and prevalence, especially in wetter environments. Since human cases often occur from infected rodents living in close proximity to humans, especially in and around buildings, this suggests that land development and rodent habitat preferences also play an important role in shaping exposure risk [49,54,55]. Our study did not investigate these hypotheses directly, but the dataset we have assembled here can contribute to future models aggregating across multiple sites and time periods.

We also note that virus taxonomy is changing in parallel with the proposed revisions to *P. maniculatus* [56]. The recent taxonomic union of Sin Nombre virus and New York virus, for example, would implicate both *P. maniculatus* and *P. leucopus* as reservoir hosts for different strains of a single virus species whose range spans from the Pacific to Atlantic coasts of North America. This viral taxonomic change is based on hierarchical genetic clustering, in part due to the high similarity of nucleoprotein and glycoprotein amino acid sequences of Sin Nombre and New York viruses. However, a human HPS

patient infected with the New York variant showed no serologic reactivity to the Sin Nombre glycoprotein, potentially indicating different seroneutralization responses in humans [57].

Further topics for research include a deeper investigation into the evidence for particular *Peromyscus* species as reservoir hosts for hantavirus and other species. Scientific definitions of reservoir host status vary significantly and prioritize different types of biological relationships and evidence, ranging from simple detection in a host to persistent pathogen maintenance with or without serious symptoms or a history of co-evolution [58–64]. Historically, hantaviruses were thought to closely co-evolve with single host species responsible for indefinitely maintaining the pathogen in the environment. Recent analyses have added nuance to this picture, showing that a number of hantavirus species infect multiple rodent hosts. More generally, biologists increasingly define reservoir hosts as composed of meta-populations or ecological assemblages of multiple species. Our results provide evidence that *P. maniculatus* sensu stricto is not the reservoir host for Sin Nombre virus in the narrow sense of being the sole biological species responsible for the pathogen's maintenance. Instead, evidence suggests that Sin Nombre virus has multiple reservoir host species, especially in light of the taxonomic union of Sin Nombre and New York viruses. More broadly, the *Peromyscus* genus is likely not monophyletic [65–67], indicating the need for an expanded survey of what is known about viral pathogens in other North American rodent species. In this respect, we lack an up-to-date, comprehensive analysis of the evidence for rodent reservoirs of hantaviruses that is consistent with leading frameworks for assessing future zoonotic disease risk [68].

Hantavirus pulmonary syndrome and other diseases spread via zoonotic viruses pose an ongoing risk to human health in the United States and surrounding areas. Rodents within the *P. maniculatus* species complex have been identified as hosts for the viruses that cause HPS and other human diseases, but the recent scientific literature on virus prevalence in this group of species has many gaps, both in terms of the types of viruses being sampled and the geographic regions in which sampling is occurring. Viral sampling has been uneven relative to the number of known human HPS cases. Taxonomic changes in the *P. maniculatus* species complex have large (and quantifiable) impacts on our knowledge of SNV prevalence and thus also HPS risk. Therefore, future work is needed to establish a systematic framework for sampling wild rodent hosts of hantaviruses, both to optimize the allocation of resources and identify regions and populations that pose disproportionate threats to human health.

## 4. Materials and methods

We conducted a systematic review of existing literature reporting observations of zoonotic viruses in the *Peromyscus maniculatus* species complex. The focal taxon was *P. maniculatus* sensu lato, and the outcomes were positive or negative test results using a range of detection methods. Examples of this include antibody tests, Polymerase Chain Reaction (PCR) tests, and genetic sequencing. Three databases, Scopus, PubMed, and PubMed Central, were used to find existing scientific literature on this topic. Search queries were conducted in August 2022 with the common and scientific names of *P. maniculatus* as well as nomenclatural synonyms and the keywords "virus", "viral", and "viruses" (Table 1). Through these parameters, 448 papers were identified as potential candidates for the systematic review.

We also added 6 ad hoc papers from other sources. Three of these [69–71] were found when we ran a supplementary search in all three databases at the reviewers' suggestion to identify any papers that used the common names "deer mice," "deermice," "deer mouse," or "deermouse." To avoid duplication of effort, we constructed these searches to return only papers that used any of these common names but did not mention "Peromyscus maniculatus." For example, the supplementary PubMed search query was: "(virus OR viral OR viruses) AND ("Deermouse" OR "Deermice" OR "Deer mouse" OR "Deer Mice") NOT ("Peromyscus maniculatus")."

We used the CADIMA web tool to review articles for inclusion [72]. We uploaded lists of identified papers from each source to CADIMA, after which we removed duplicates, resulting in 253 remaining papers. Each paper was manually reviewed by AF for inclusion, and a second person (BS) was consulted on unclear cases. Inclusion criteria were that the

**Table 1. Search terms used in conducting the systematic review of existing literature across three separate databases. Common names as well as scientific names for species within the _P. maniculatus_ species complex were included.**

| Database | Search Query Used | Search Settings Used |
|---|---|---|
| PubMed | (virus OR viral OR viruses) AND ("Peromyscus maniculatus" OR "Peromyscus gracilis" OR "Peromyscus bairdi" OR "Peromyscus abietorium" OR "Peromyscus nubiterrae" OR "Peromyscus argentatus" OR "Peromyscus eremus" OR "Peromyscus anticostiensis" OR "Peromyscus plumbeus" OR "Peromyscus gambelii" OR "Peromyscus labecula" OR "Peromyscus arcticus" OR "Peromyscus sonoriensis" OR "Eastern Deermouse" OR "North American Deermouse") | Title/Abstract |
| PubMed Central | | Abstract OR Body-All Words OR Title |
| Scopus | "Peromyscus maniculatus" OR "Peromyscus gracilis" OR "Peromyscus bairdi" OR "Peromyscus abietorium" OR "Peromyscus nubiterrae" OR "Peromyscus argentatus" OR "Peromyscus eremus" OR "Peromyscus anticostiensis" OR "Peromyscus plumbeus" OR "Peromyscus gambelii" OR "Peromyscus labecula" OR "Peromyscus arcticus" OR "Peromyscus sonoriensis" OR "Eastern Deermouse" OR "North American Deermouse" AND virus OR viral OR viruses | TITLE-ABSTRACT-KEY |

article must report new primary data about the results of testing wild _P. maniculatus_ for virus occurrence or infection. For reasons of scope, we excluded lab studies based on artificial infection experiments as well as papers published prior to 2000. We excluded 192 papers for one or more of the following reasons: not having a full text available in English through interlibrary loan or subscription services at the author's home institution, not meeting the inclusion criteria, or being published before the cutoff period. This left 49 papers for data extraction from the systematic review process (Fig 1). The full list of included and excluded studies is available in S2 Table.

Data extraction entailed reading each scientific paper in full and extracting relevant information on sampling time, sampling location, number of hosts, various host identifiers, detection method used, material sampled, and the results of observations into a spreadsheet (S4 Table). Information on all species captured and sampled was recorded for future use, though this study focuses solely on detection presented in _P. maniculatus_ (S3 Table). Summary results for _P. maniculatus_ are available in S5 Table.

To visually represent the patterns discovered in the extracted data, we created a series of maps using the Tableau software program [73]. Spatially coded results were taken from the data extraction spreadsheet and translated into the maps on either the county, province, or state level, depending on whether the sample was taken in the United States, Canada, or Mexico, respectively.

Sampling effort was represented by the number of observations, which are defined as the number of times a unique host was tested for a unique virus using a specific method. Hence, if an individual rodent was tested by ELISA for both Whitewater Arroyo virus (WWAV) and Amapari virus (AMAV), this was recorded as two separate observations. Similarly, there would be multiple observations reported if an individual rodent was tested for the same virus multiple times in a recapture study. This way of individuating observations therefore provides a more fine-grained basis for collecting and analyzing test results. In the case of counties that were the site of multiple studies, the observations across each study were summed to create a single value which was then reflected in the map.

In the rare instance where one county was sampled for multiple virus species, the higher-order classification of virus species was used to create the maps. For example, if one study reported sampling for SNV and a separate study conducted in the same county tested for hantavirus antibodies but not for a specific virus species within that family, the county was represented as a hantavirus county on the map. However, all results for each type of virus tested are available in S4 Table.

Virus prevalence was determined as the percentage of antibody-positive _P. maniculatus_ reported by each study. For sampling locales with multiple studies and thus multiple prevalence results to consider, the reported prevalence results were taken and weighted by sampling effort in order to avoid over-representing results from studies that tested a smaller number of rodents. The county estimates were calculated according to the following equation:

$$P_c = \sum w_s \times p_s$$

The weights $w_s$ are the number of rodents tested for that pathogen in each study divided by the sum of all rodents tested. Thus, the weights relative to a county sum to 1. The weight for each study is multiplied by the seroprevalence proportion $p_s$ from each study, calculated as the number of positive tests divided by the total [74]. Weighting is important to account for the relative precision of different studies in estimating the mean prevalence of the pathogen in the host population.

To assess the richness and distribution of the viruses detected across the selected literature, we processed the data into cumulative measures of sampling effort by virus type and geographic location. Geographic regions were analyzed at the county level for the United States and the state or province level for any other sampling locales. To track how sampling effort has been distributed over time, the relationship between the number of observations reported and (a) the number of viruses identified and (b) the number of unique locations sampled were also examined.

To assess the feasibility of accurately distinguishing SNV from other hantaviruses using DNA sequence data, we downloaded 79 sequences of hantavirus with *Peromyscus maniculatus* as the host from the ZOVER database on 4 Nov 2024. We then aligned those empirical sequences to verified NCBI RefSeq sequences from 28 hantavirus species (downloaded 8 Nov 2024), separately partitioning the M, S, and L segments for alignment. Sequences from El Moro Canyon virus were subsequently added to the S-segment phylogeny to assess its phylogenetic placement related to SNV (downloaded 17 Mar 2025). Alignments were made using MAFFT v7.490 software [75] and maximum likelihood trees were constructed using RAxML v8.2.12 [76], specifying the GAMMA+CAT model performing 100 bootstrap replicates.

## Supporting information

**S1 PRISMA Checklist. Completed checklist for systematic reviews following the 2020 PRISMA Standard [77].** (DOCX)

**S1 Fig. Phylogeny of the hantavirus S genome segment based on a maximum-likelihood alignment of hantavirus nucleotide sequences collected from *Peromyscus maniculatus*.** The sequence label in red is the Sin Nombre reference sequence from NCBI. Sequences highlighted in pink are El Moro Canyon virus (*Orthohantavirus carrizalense*) collected in 1996 by Rawlings et al. [36]. *P. maniculatus* sequences highlighted in blue are used for orientation across the S, M, and L segments, since these sequences are all derived from a single study [37]. (PDF)

**S2 Fig. Phylogeny of the hantavirus M genome segment based on a maximum-likelihood alignment of hantavirus nucleotide sequences collected from Peromyscus maniculatus.** The sequence label in red is the Sin Nombre reference sequence from NCBI. P. maniculatus sequences highlighted in blue are used for orientation across the S, M, and L segments, since these sequences are all derived from a single study [37]. (PDF)

**S3 Fig. Phylogeny of the hantavirus L genome segment based on a maximum-likelihood alignment of hantavirus nucleotide sequences collected from Peromyscus maniculatus.** The sequence label in red is the Sin Nombre reference sequence from NCBI. P. maniculatus sequences highlighted in blue are used for orientation across the S, M, and L segments, since these sequences are all derived from a single study [37]. (PDF)

**S1 Table. Summary of search queries and results from Scopus, PubMed, and PubMed Central databases.** The first three search queries show how results depend on the use of scientific and common name synonyms for *P.*

*maniculatus*. The final search query shows the number of results from each database used as inputs for the systematic review process.
(XLSX)

**S2 Table. Numbered list of included and excluded studies from the systematic review.**
(XLSX)

**S3 Table. Summary of sampling locations, effort, seroprevalence, detection method, and target virus in each study included in the review.**
(XLSX)

**S4 Table. Comprehensive table of all data extracted from the included studies.**
(XLSX)

**S5 Table. Intermediate results for calculating weighted prevalences at the county level in the U.S. and province level in Canada and Mexico.** Summary statistics of human HPS cases and rodent and virus sampling efforts.
(XLSX)

## Acknowledgments

Our thanks to DeeAnn Reeder, Jorrit Poelen, and Donat Agosti for sharing their insights and suggestions on data extraction and synthesis methods. We'd also like to recognize several other Arizona State University undergraduate students who have assisted this project along the way, including Linnea Donovan, Gwen Calaro, and Tanishq Jain.

## Author contributions

**Conceptualization:** Ally Finkbeiner, Nathan Upham, Beckett Sterner.

**Data curation:** Ally Finkbeiner, Beckett Sterner.

**Formal analysis:** Ally Finkbeiner, Nathan Upham, Beckett Sterner.

**Funding acquisition:** Nathan Upham, Beckett Sterner.

**Investigation:** Ally Finkbeiner, Ahmad Khatib, Beckett Sterner.

**Methodology:** Beckett Sterner.

**Project administration:** Beckett Sterner.

**Supervision:** Beckett Sterner.

**Visualization:** Ahmad Khatib, Beckett Sterner.

**Writing – original draft:** Ally Finkbeiner, Beckett Sterner.

**Writing – review & editing:** Ally Finkbeiner, Ahmad Khatib, Nathan Upham, Beckett Sterner.

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
