## [Decision Letter · Decision Letter 0]

18 Sep 2024

Dear Dr. Sterner,

Thank you very much for submitting your manuscript "A Systematic Review of the Distribution and Prevalence of Viruses Detected in the Peromyscus maniculatus Species Complex (Rodentia: Cricetidae)" for consideration at PLOS Pathogens. As with all papers reviewed by the journal, your manuscript was reviewed by members of the editorial board and by several independent reviewers. In light of the reviews (below this email), we would like to invite the resubmission of a significantly-revised version that takes into account the reviewers' comments.

Based on especially Reviewer 3's comments, I'd strongly advise that the authors double-check their searches thoroughly using both "deer mouse", "deer mice", "deermouse" and "deer mice" in addition to "Peromyscus" to ensure both accurate literature citing and scoping. As a side note, in response to reviewers: HPS, not HCPS, is the official term for the disease as per the WHO ICD-11, so I would recommend leaving this; likewise, "deermouse/deermice" (no space) is the correct term for all animals classified in genus _Permomyscus_ per the zoological reference standard (Wilson & Reeder).

We cannot make any decision about publication until we have seen the revised manuscript and your response to the reviewers' comments. Your revised manuscript is also likely to be sent to reviewers for further evaluation.

Sincerely,

Jens H. Kuhn

Academic Editor

PLOS Pathogens

Ronald Swanstrom

Section Editor

PLOS Pathogens

Michael Malim

Editor-in-Chief

PLOS Pathogens

orcid.org/0000-0002-7699-2064

Reviewer's Responses to Questions

**Part I - Summary**

Reviewer #1: The manuscript by Finkbeiner et al. conducted a systematic literature review to examine the distribution of viruses in North American deer mice, which has recently been proposed to be split into different species. The work reviewed the literature and conducted an analysis to identify the distribution of viruses, principally hantaviruses, arenaviruses and flaviviruses, amongst the proposed new peromyscus species. This work provides clarity of viral distributions and provides a compelling argument that the proposed P. sonoriensis is the principal reservoir host of Sin Nombre virus but that the virus can also be found in other species. Complicating this matter is the efforts by the ICTV to rework taxonomy of some of these viruses, such as the merging of SNV with NY1 hantavirus, which is hosted by P. leucopus. The manuscript is well written and the efforts of the authors are admirable.

Reviewer #2: The manuscript by Finkbeiner et al. describes a systematic review conducted by the authors, in which they extracted data from all relevant papers looking at natural viral infection of Peromyscus since 2000. Since deer mice are a critical reservoir for SNV and other viruses (not to mention pathogenic bacteria and parasites), this work is a timely description of what studies have been done to date surveying infection of these mice. The authors describe weighted seroprevalence mostly for SNV and describe the strong bias toward sampling in the western United States compared to other regions. They also do a nice job discussing the complexity of how to group together host species and viruses given the recent suggested taxonomic update of the Peromyscus genus, which as they note, means that 92% of described infections need to be reclassified. The manuscript nicely distills information from 46 studies and >60,000 observations into one well-written and well-presented package. I have some comments and questions regarding a few of the authors’ choices.

Reviewer #3: The authors present a review of Peromyscus-borne viral pathogens based on literature searches. The approach seems sound however details pertaining to virology seem weak and in many cases are misleading. Several references are missing which may suggest flaws in the search terms. For example, Goodfellow et al. 2021 (PMID 34549977) appears to be absent from the listed references, yet it describes similar concerns with the current knowledge of hantavirus / rodent reservoir theories. In my brief searches, this is the third paper that comes up if one searches for "Sin Nombre virus AND deer mice" .Further, there is no mention of long-term ecology / prevalence studies conducted across the USA post-1993 attempting to define indicators for predicting hantavirus spillover events.

**Part II – Major Issues: Key Experiments Required for Acceptance**

Reviewer #1: It would have been helpful if the authors had included other viruses hosted by peromyscus rodents, such as Modoc virus and El Moro Canyon virus (which is also found principally in western harvest mice). In the end, the viruses/reservoirs are somewhat "fuzzy" simply because different viruses can be found in multiple species.

The authors should clarify how the serology was done in these studies (e.g., "primary antigen" page 8). There is a huge difference between ELISA with nucleocapsid (highly conserved among New World hantaviruses) and Gn/Gc (more divergent), and assays such as focus reduction tests with cell culture/infectious virus. Antibodies to SNV nucleocapsid are cross reactive with Andes virus nucleocapsid, for example. This can have a profound impact on the analyses and conclusions raised by the authors. This should also be discussed in further detail in the Discussion on page 13 ("virus-specific antibody" tests).

Reviewer #2: (No Response)

Reviewer #3: - As stated above the description of virology in this paper needs to be improved. The basic description of virus detection methods lead to confusion and are potentially misleading. Hantaviruses are known to be serologically cross reactive and most studies conducted to date relay on basic ELSIA methodologies to determine the prevalence of hantaviruses in rodents. To determine it is SNV with certainty requires molecular techniques and often sequence analysis, which was frequently not done in earlier studies. Although serotyping ELISAs have been described, they are not commonly used therefore statements like "...studies are focused on SNV antibodies.." misleading. The studies are focused on antibodies reactive to hantaviruses. Further, the statement "Results on Powassan virus, Whitewater Arroyo Virus, Amapari virus and Mononhahela virus are inbformative in that they indicate the presence of these other hantaviruses within the P. maniculatus complex" is incorrect. Most of the viruses listed are not hantaviruses. This statement further highlights the incomplete or inaccurate knowledge of viruses presented in this manuscript.

**Part III – Minor Issues: Editorial and Data Presentation Modifications**

Reviewer #1: Abstract. Although originally thought to be a pulmonary disease, SNV causes a cardiopulmonary disease. This is why clinical efforts are principally focused on the cardiovascular system. This should be added to the abstract (and why most of us refer to it has HCPS).

Also in the abstract, do not capitalize "hantavirus (cardio)pulmonary syndrome" because it is not a proper noun. Again on page 5.

Author Summary. What is meant by "locked"? Also, if referring to the families, then use Hantaviridae and Arenaviridae and italicize.

Change last sentence to "major knowledge gaps that remain."

Introduction. Use "deermouse" or "deer mouse" (e.g., second line of page 7), but not both. Do not capitalized "deermouse" nor "Tick-borne" because they are not a proper nouns. I also suggest the authors add "deer mouse" to the keyword list; otherwise, the risk is the paper will not be found by those who search for "deer mouse."

Change "mortality rate" to "case fatality rate." They are different terms.

Figure 3 is out of order. The figures should be in the order they are referenced in the text. Figure 3 should be the last figure.

Reviewer #2: I am wondering why the authors chose to specifically include “viruses detected in Peromyscus” in the title, rather than simply a scoping review on SNV? As it stands, the arenaviruses and Powassan make up such a minority of the data that this might have made more sense, with just a mention of the other viruses, which is basically how it is already.

In the results section starting with “other viruses were sampled less frequently…” it discusses 1 study with 353 observations for AMAV and WWAV. The mentioned of “arenavirus” tested for in only a single study, while then mentioning multiple studies later on looking at AMAV and WWAV is a bit confusing, even though I know they mean a broader Arenavirus positivity and not one that is species specific. To this end, in the last sentence of this paragraph it also says that there are multiple studies focusing on arenaviruses including WWAV and AMAV, but I think this adds to the confusion without clarifying the difference, because in the preceding paragraph is says there are only 2 studies looking at WWAV and AMAV. I would maybe make this distinction a bit more clear.

Minor comments:

Discussion; sentence starting with “results on Powassan virus,…” it says they are informative as the indicate the presence of these other hantaviruses, though these are not all hantaviruses. And perhaps a brief discussion about Monongahela virus and how it is not its own species as of the most recent taxonomic update might be warranted.

There is a section of the discussion where the authors discuss variability in seroprevalence and how this is likely due to inconsistencies in sampling across locations and timing. It may be worth a brief discussion on what is known about how rates of SNV prevalence change in deer mouse populations depending on season and how the timing of individuals studies might influence the data in this regard. In lieu of not including timing of sampling in individual studies as part of the wider analysis

Reviewer #3: - Authors include Eastern deermouse and North American deermouse in search terms but never simply deer mouse. I get zero results for North American deermouse but get 3,236 hits for deer mouse on Pubmed. Perhaps deer mouse should be included in the search criteria? Is it deermouse or deer mouse? I more commonly see it as deer mouse, yet the search terms used list "deermouse"

- In the introduction the authors should differentiate between studies demonstrating susceptibility (which I assume is the term the authors use for experimental infections) versus known to harbor which appears to be based on ecological studies. For example, they cite a study by "Griffin" in 2021 (which appears to be an incomplete reference) suggesting deer mice are susceptible to SARS-Cov-2 but as far as this reviewer is aware deer mice have not been implicated as natural reservoirs for this virus.

-I would encourage the authors to further analyze their datasets to highlight which studies conducted in-depth rodent identification techniques such as cytochrome sequencing to determine rodent species as opposed to those which simply used visual clues.

- Many critical references are missing which makes this reviewer wonder if the search terms used need to be expanded. To this point, there are several papers from studies conducted in Canada that seem to have been missed by the terms used.

PLOS authors have the option to publish the peer review history of their article (what does this mean? ). If published, this will include your full peer review and any attached files.

**Do you want your identity to be public for this peer review?** For information about this choice, including consent withdrawal, please see our Privacy Policy .

Reviewer #1: No

Reviewer #2: No

Reviewer #3: No
---

## [Decision Letter · Decision Letter 1]

6 Feb 2025

PPATHOGENS-D-24-01530R1

A Systematic Review of the Distribution and Prevalence of Viruses Detected in the Peromyscus maniculatus Species Complex (Rodentia: Cricetidae)

PLOS Pathogens

Dear Dr. Sterner,

Thank you for submitting your manuscript to PLOS Pathogens. After careful consideration, we feel that it has merit but does not fully meet PLOS Pathogens's publication criteria as it currently stands. Therefore, we invite you to submit a revised version of the manuscript that addresses the points raised during the review process.

Please submit your revised manuscript within 30 days Apr 07 2025 11:59PM. If you will need more time than this to complete your revisions, please reply to this message or contact the journal office at plospathogens@plos.org. Please include the following items when submitting your revised manuscript:

We look forward to receiving your revised manuscript.

Kind regards,

Ronald Swanstrom

Section Editor

PLOS Pathogens

Ronald Swanstrom

Section Editor

PLOS Pathogens

Sumita Bhaduri-McIntosh

Editor-in-Chief

PLOS Pathogens

orcid.org/0000-0003-2946-9497

Michael Malim

Editor-in-Chief

PLOS Pathogens

orcid.org/0000-0002-7699-2064

**Journal Requirements:**

1) Please upload a copy of Figures 1-7 which you refer to in your text. Please note that the figures should be uploaded as separate Figure files in .tif or .eps format. For more information about how to convert and format your figure files please see our guidelines: 

2) We have noticed that you have uploaded Supporting Information files, but you have not included a complete list of legends. Please add a full list of legends for the supplementary tables after the references list.

3) For maps included in the manuscript: Please (a) provide a direct link to the base layer of the map (i.e., the country or region border shape) and ensure this is also included in the figure legend; and (b) provide a link to the terms of use / license information for the base layer image or shapefile. We cannot publish proprietary or copyrighted maps (e.g. Google Maps, Mapquest) and the terms of use for your map base layer must be compatible with our CC BY 4.0 license.

4) All authors should have affiliations, and no affiliations should be included unless linked to an author. Please ensure that the affiliations of Nathan Upham and Beckett Sterner are provided in the title page. Please also ensure that the affiliation of  Beckett Sterner is included in the online submission form.

5) As required by our policy on Data Availability, please ensure your manuscript or supplementary information includes the following:

**Reviewers' Comments:**

Reviewer's Responses to Questions

**Part I - Summary**

Reviewer #1: The authors have addressed the issues from the previous manuscript but have not addressed El Moro Canyon virus that has been detected in deer mice (see PMID 9025697). It is not clear that other hantaviruses might infect deer mice, thus the assertion that seropositive deer mice must be infected with SNV is not justified.

Reviewer #2: The authors have addressed my comments and those of the other reviewers and editor

Reviewer #3: (No Response)

**Part II – Major Issues: Key Experiments Required for Acceptance**

Reviewer #1: There are no experiments; this is an analysis of the literature.

Reviewer #2: None.

Reviewer #3: (No Response)

**Part III – Minor Issues: Editorial and Data Presentation Modifications**

Reviewer #1: None.

Reviewer #2: (No Response)

Reviewer #3: The authors have addressed my main concerns in their revised submission. I disagree with the authors responses that the years ecological work conducted on SNV and other hantaviruses are out of scope for this work. I would prefer to see some further discussion on that topic since those studies represent a vast amount of knowledge that is in my opinion highly relevant to this topic.

PLOS authors have the option to publish the peer review history of their article (what does this mean? ). If published, this will include your full peer review and any attached files.

**Do you want your identity to be public for this peer review?** For information about this choice, including consent withdrawal, please see our Privacy Policy .

Reviewer #1: No

Reviewer #2: No

Reviewer #3: No

**Figure resubmission:**
---

## [Editor Report · Decision Letter 2]

11 Apr 2025

Dear Dr. Sterner,

We are pleased to inform you that your manuscript 'A Systematic Review of the Distribution and Prevalence of Viruses Detected in the Peromyscus maniculatus Species Complex (Rodentia: Cricetidae)' has been provisionally accepted for publication in PLOS Pathogens.

Best regards,

Jens H. Kuhn

Academic Editor

PLOS Pathogens

Ronald Swanstrom

Section Editor

PLOS Pathogens

Sumita Bhaduri-McIntosh

Editor-in-Chief

PLOS Pathogens

orcid.org/0000-0003-2946-9497

Michael Malim

Editor-in-Chief

PLOS Pathogens

orcid.org/0000-0002-7699-2064
---

## [Editor Report · Acceptance letter]

Dear Dr. Sterner,

We are delighted to inform you that your manuscript, "A Systematic Review of the Distribution and Prevalence of Viruses Detected in the *Peromyscus maniculatus* Species Complex (Rodentia: Cricetidae)," has been formally accepted for publication in PLOS Pathogens.

Best regards,

Sumita Bhaduri-McIntosh

Editor-in-Chief

PLOS Pathogens

orcid.org/0000-0003-2946-9497

Michael Malim

Editor-in-Chief

PLOS Pathogens

orcid.org/0000-0002-7699-2064